# Deliberative and Affective Risky Decisions in Teenagers: Different Associations with Maladaptive Psychological Functioning and Difficulties in Emotion Regulation?

**DOI:** 10.3390/children9121915

**Published:** 2022-12-07

**Authors:** Marco Lauriola, Luca Cerniglia, Renata Tambelli, Silvia Cimino

**Affiliations:** 1Department of Social and Developmental Psychology, Sapienza, University of Rome, 00185 Roma, Italy; 2Faculty of Psychology, International Telematic University Uninettuno, 00186 Roma, Italy; 3Department of Dynamic, Clinical and Health Psychology, Sapienza, University of Rome, 00185 Roma, Italy

**Keywords:** deliberative and affective processes, risk propensity, Columbia Card Task, youth self-report, difficulties in emotion regulation, thought problems, executive functions, network analysis

## Abstract

Using network analysis, we investigated the relationships between maladaptive psychological functioning, difficulties in emotion regulation, and risk-taking in deliberative and affective behavioral decisions. Participants (103 adolescents aged between 13 and 19 years, 62% boys) took the Cold (deliberative) and Hot (affective) versions of the Columbia Card Task and completed the Youth Self-Report (YSR) and the Difficulties in Emotion Regulation Scale (DERS). In contrast to the view that risk propensity increases from preadolescence to middle adolescence and decreases at later ages, our study revealed no age-specific trend. YSR syndrome scales were significantly correlated with risk propensity, but only in the Cold version. The YSR Thought Problems scale was the most central node in the network, linking internalizing and externalizing problems with risk propensity in the Cold CCT. Lack of emotional Clarity was the only DERS consistently linked with risk-taking both in correlation and network analyses. Maladaptive psychological functioning and difficulties in emotion regulation were linked with risk propensity in affective risky decisions through deliberative processes. The statistical significance of direct and indirect effects was further examined using nonparametric mediation analyses. Our study highlights the role of cognitive factors that in each variable set might account for risk-taking in teenagers.

## 1. Introduction

In recent decades, researchers in clinical and developmental psychology have turned their attention to the neurocognitive factors underlying decision-making and emotion regulation to understand risk-taking and real-world psychopathology [1,2,3]. The first reason driving this interest is that the brain networks responsible for adaptive decision-making and emotion regulation are often impaired in individuals suffering from mental illnesses (e.g., [4,5,6,7,8]). Second, mental disorders and problem behaviors with potentially harmful consequences surge during adolescence, when the abovementioned networks are still developing [9,10,11]. Third, although risk-taking is normative in teenagers [12,13], it might predict psychological problems, such as conduct disorder and depression (e.g., [14]), substance use, and addiction (e.g., [15]). Collectively, the literature suggests that mental health, emotion regulation, and decision-making are intertwined in youth. The present study investigates these relationships, using scales with comprehensive coverage of behavioral-emotional problems and difficulties in emotion regulation, distinguishing risk propensity in decision tasks tapping into “cold” and “hot” executive functions.

Executive functions are higher-order skills needed for logical reasoning, planned behavior, emotion regulation, and social functioning [16]. Cognitive flexibility, working memory, and response inhibition, to name a few, are cold functions depending on the brain activity in the dorsolateral prefrontal cortex (DLPFC) and are measured using decontextualized, abstract tasks [17]. Decision-making and emotion regulation are hot functions that hinge upon the ventral and medial areas of the prefrontal cortex (VMPFC) and are deployed in situations where motivational drives and emotions are salient [17]. The difference between hot and cold functions is somewhat oversimplified, and it is not entirely clear whether and which context or scenario prompts hot or cold processes [18]. Nevertheless, research using cold and hot tasks has shown that executive functions develop according to separable yet woven trajectories [19].

Although developmental trends change slightly depending on specific tasks and abilities, cold functions typically reach an adult-like performance level on simple tasks by age 12; however, using more demanding tasks and assessing certain functions (e.g., visual-spatial working memory, inhibitory control under higher cognitive load) suggests that cold functions continue to develop beyond 12 years [20]. In support of this claim, cross-sectional studies highlighted the improvement in cold functions (e.g., working memory and planning) throughout adolescence [21]. Moreover, response inhibition was found to be still underdeveloped in early and middle adolescence [22]. Due to task heterogeneity and fewer longitudinal studies, developmental trends are less clear for hot functions.

According to Zelazo and Carlson [19], the Iowa Gambling Task (IGT) is a good marker of hot functions, given the emotional salience of penalties and rewards and task sensitivity to neurocognitive deficits, such as lesions in the VMPFC [23,24]. Regarding developmental trajectories, Almy and colleagues [25] found a linear improvement with age in IGT performance, from childhood to late adolescence. However, three cross-sectional studies found a quadratic trend, diverging on whether the minimum performance is at 9 [26], 12 [27], or 15 [28] years. These latter studies agree that hot functions develop not linearly, maturing faster in late adolescence or young adulthood.

Because Steinberg (2010) also assessed the cold functions, which increased linearly with age, the results were interpreted based on the interaction between changes in a socioemotional system (located in the limbic and paralimbic brain regions) and a cognitive control system (located in the lateral prefrontal and parietal cortices) [29,30]. Accordingly, adolescent risk-taking results from increased dopaminergic activity within the socioemotional system at puberty, leading to a rise in reward-seeking, not compensated by the maturation of executive functions, responsible for self-regulation and impulse control [12]. According to this theory, adolescents are thought to be more likely than children and adults to make risky decisions in affective contexts where emotions are at play, although there may be differences from individual to individual and developmental stages in the level of emotional arousal evoked by different contexts [18,31].

Executive functions have also been proposed as transdiagnostic mechanisms for mental disorders in youth [32]. Not only is it widely accepted that internalizing problems are linked to poor performance on cold tasks, but brain imaging studies have also validated this association (e.g., [33]). Similarly, the correlation between executive function deficits and externalizing problems is well established (e.g., [34]). A debated question is whether deficits in executive functions precede or follow the onset of mental problems in adolescence. Two longitudinal studies supported the latter claim. For example, a higher frequency of internalizing and externalizing symptoms at 13–14 years predicted lower performance on cold tasks at 17–18 years of age, but not the other way around [35]. Likewise, Morea and Calvete [36] studied a sample of adolescents aged 12 to 17 years for a year, showing that internalizing problems predicted subsequent deficits in executive functions.

The success of the IGT as a marker of hot executive functions and its widespread use in the neuropsychological assessment of individuals diagnosed with mental illnesses (for a review, see [37] has inspired the development of new behavioral tasks designed to measure decision-making processes in greater detail [38,39]. The Columbia Card Task (CCT) was developed to assess risk propensity in hot (affective) and cold (deliberative) decision contexts [40]. The hot CCT requires stepwise incremental decisions and provides immediate feedback to participants; the cold CCT does not provide immediate feedback, and participants only see the game’s final result. Compared to cold CCT, hot CCT was found to trigger powerful emotional arousal [41,42]. Conversely, the cold CCT triggers more deliberative decision-making.

Risk propensity and use of information in the cold CCT were found to be correlated with mental flexibility [43], cognitive control and analytic thinking [44], attention, and working memory [40]. However, penalties and rewards are still part of the cold CCT, and choices are motivated to maximize gains (see Instruments below). This feature makes the cold CCT not completely devoid of emotions. Rather, emotions are experienced more abstractly. For example, the decision-maker can anticipate and consider emotions that are expected to arise because of choice outcomes (e.g., how one might feel if things go awry) [45]. These anticipated emotions can assist the decision-maker in regulating present behavior (e.g., [46]). By contrast, the hot CCT involves integral emotions reflecting the sense of escalating tension and excitement associated with the upcoming feedback.

The primary purpose of the present study is to investigate the relationships among three sets of variables that the literature suggests may underlie the surge in risk-taking during adolescence. These are maladaptive psychological functioning, difficulties in emotion regulation, and risk propensity in affective and deliberative decision tasks. Using a network analysis approach, we first seek to map the overall structure of the relationships between the sets of variables. Second, we aim to identify the network’s most central elements (nodes) in each set as they may suggest key variables in risky adolescent decisions and potential targets for psychological intervention [47,48]. Finally, we intended to test whether connections between network elements (edges) might suggest direct and indirect connections between sets. In general, we hypothesized greater maladaptive psychological functioning to be related to increased risk propensity. Second, we expected emotion regulation difficulties to be associated with maladaptive psychological functioning and risk propensity. Third, we explore whether the abovementioned variables have different associations with risk propensity under deliberative and affective decision-making conditions.

## 2. Materials and Methods

### 2.1. Participants

A total of 103 Italian adolescents (39 girls and 66 boys) were randomly recruited from a secondary school in the city of Rome. There were no exclusion criteria. The study was first presented to the school headmaster and subsequently to the parents of eligible participants from which written informed consent was collected (except if the participant’s age was over 18 years in which case informed consent was collected from the participants themselves). Participants’ ages varied between 13 and 19 years (M  =  15.56; SD  =  1.19). No statistically significant difference between boys and girls was found for age (t-value  =  0.27; *p*  =  0.788). Most of the subjects were Caucasian (97.9 percent), and 89 percent of their families had a household income between €28,000 and €55,000 per year (corresponding to an average socioeconomic status, SES). Among adolescents. 89.9% belonged to intact family groups. The local ethical committee for psychological research approved the study (Prot. n. 0000018). The measures were administered in two sessions spaced one week apart. In session one, the participants completed the self-report scales (description see below); in session two, the Columbia Card Task (description see below) was administered in the school’s computer room. The hot and cold versions were presented in randomized order. The study’s data are provided as electronic Appendix A.

### 2.2. Instruments

#### 2.2.1. Columbia Card Task

We used abbreviated versions of the hot and cold CCT, each consisting of 27 trials instead of the 54 trials used in the original task [40]. In the cold task, each trial showed 32 cards, face down, arranged in four rows of eight cards each. Participants are asked to choose how many cards they want to turn over by indicating a number between 0 and 32 at the top of the screen, and each flipped card provides some gain. Each trial began with a zero score. Participants were told that they could lose a certain number of points if they encountered a penalty card within the number of cards they had decided to turn over. The goal was to score as many points as possible, and participants were informed (in the upper right corner of the screen) about potential gains (i.e., +10, +20, and +30 points per card), potential losses (i.e., −250, −500, and −750 points if a penalty card is encountered), and penalty cards (i.e., 1, 2, and 3). Consistent with the original task, the 27 trials represented an orthogonal combination of the factors mentioned above. The display still shows 32 cards in the hot task, and the game factors are the same as in the cold task. However, participants can sequentially turn over cards and stop at any time, collecting the points obtained up to that moment. In the original hot CCT, the losing cards were always the last cards on display, making this task somewhat rigged. To prevent adolescents from realizing this feature and revealing information to schoolmates, the cards were shuffled in each trial, and the losing card could turn up at any point during the task according to an increasing probability function. Previous research has shown that risk-taking tendencies can be reliably assessed using the “unrigged” version of the task [41,42]. Because flipping more cards confer greater potential gains and a higher risk of encountering a penalty card, the primary index of the CCT was the number of flipped cards on each trial (Flipped Cards, FC), with higher scores indicating greater risk-taking.

#### 2.2.2. Youth Self-Report (YSR)

Youth self-report (YSR) (Achenbach, 1991; Italian version—Frigerio et al., 2001) is a self-report questionnaire that covers behavioral and emotional problems in the past 6 months. It contains 112 problem items, which are scored on a 3-point scale (0 = not true, 1 = somewhat or sometimes true, 2 = very or often true). The YSR total problem scale can be divided into syndrome subscales: Anxious/depressed, Withdrawn, Somatization, Social problems, Thought problems, Attention problems, Rule breaking, and Aggression. Withdrawn, Somatization, and Anxious/Depressed together comprise a broad “Internalizing” dimension (31 items), whereas Rule-breaking and Aggressive behaviors together constitute an “Externalizing” dimension (32 items; Achenbach, 1991). Higher scores on these scales indicate more maladaptive functioning. Some YSR items are included in the “Other problems” subscale (32 items). Achenbach and Rescorla (2001) found that the internal for the empirically based problem scales was supported by Cronbach’s alphas ranging from 0.71 to 0.95.

#### 2.2.3. Difficulties in Emotion Regulation Strategies (DERS)

The Difficulties in Emotion Regulation Strategies [49] is a self-report questionnaire comprised of 36 Likert-type items arranged into six subscales. It assesses: (1) the lack of acceptance of the emotional responses (i.e., non-acceptance), (2) difficulty in controlling impulsive behaviors and behaving in accordance with desired goals (i.e., goals), (3) limited access to emotion regulation strategies (i.e., strategies), (4) lack of control when experiencing intense emotions (i.e., impulse), (5) difficulties in recognizing emotions (i.e., clarity), and (6) limited awareness and understanding of emotions (i.e., awareness”). Higher scores in each subscale reflect more emotion regulation difficulties. (i.e., accepting emotional reactions to distress, being able to stay focused on and accomplishing tasks when distressed, controlling immediate responses to being able to regulate emotions when upset).

### 2.3. Data Analysis

#### 2.3.1. Preliminary Analysis

Total scores were calculated for each of the study variables (Table 1). The Shapiro–Wilk’s test was significant for all variables (all *p*-s < 0.001) suggesting violations of normality. The data distribution was moderately asymmetrical (skewness between −1.0 and 1.0) for all variables and raised some concern only for Social problems, Thought problems, and Rule-breaking (skewness equal to 1.43, 1.12, and 1.20, respectively). The Kurtosis was between −3 and 3 for all variables, indicating that extreme values were not very different from those expected according to normal data distribution. To cope with violations of normality, a non-paranormal transformation of the variables was applied before network analysis [50]. Nonparametric Spearman correlations were used to check significant associations among study variables.

#### 2.3.2. Network Analysis

We used the bootnet and qgraph packages for R to estimate, test, and visualize a network model based on Youth Self-Report syndrome scales, Difficulties in Emotion Regulation Strategies, and CCT task scores. Given the small number of participants relative to the number of variables in the network, we applied the EBIC-Glasso estimation method, which returns a parsimonious network model with the smallest number of parameters explaining the covariation structure of the data. Under EBIC-Glasso estimation, nonzero edges connecting nodes can be interpreted as partial correlations between variables, controlling for all other variables in the network. The edge thickness represents the strength of an association. In addition to visual inspection, the importance of nodes in the network was assessed using centrality indices. Strength centrality indicates the number of edges a node has with other nodes and represents the likelihood that a given node can activate other nodes in the network. The degree to which a node is in-between others and its importance in terms of interconnection is measured using betweenness centrality. Closeness centrality is related to the average distance of a node from others in the network and represents the level of dependence of a node from other nodes.

One crucial feature of a network model is stability, namely, the tendency of the network structure to resist change as research participants are progressively excluded from the sample. According to Epskamp et al. [51], the correlation stability coefficient is an index of network stability, calculated as the correlation between centrality indices resulting from the original network and those resulting from a network estimated on a subset of cases (usually 70%). The correlation stability coefficient should not be lower than 0.25 and preferably above 0.5 to ensure network stability. A second desirable property of a network structure is edge accuracy, that is the extent to which the connections between nodes can be considered reliable. Taking a nonparametric approach edge accuracy was estimated using 95% bootstrap CIs. A noteworthy point is that non-zero edges in the network are already strong enough to be included in the model as they survived the Glasso regularization, hence the confidence intervals should not be meant as statistical tests of significance.

#### 2.3.3. Mediation Analyses

We used the *lavaan* package for R to test hypotheses concerning the direct and indirect relationships emerging from Network Analysis. The model’s fit was primarily assessed using the Comparative Fit Index (CFI), Tucker-Lewis Index (TLI), Root Mean Square Error of Approximation (RMSEA), and Standardized Root Mean Square Residual (SRMR). CFI and TLI greater than 0.95 indicate a good fit, with values above 0.90 deemed acceptable. A good fit is also supported by RMSEA and SRMR lower than 0.06 and 0.08, respectively. The significance of indirect effects was tested using bootstrap 95% confidence intervals with 5000 resamplings. Values around 0.02, 0.13, and 0.26 represent small, medium, and large effect size thresholds for standardized indirect effects [52].

## 3. Results

### 3.1. Descriptive Analysis

Figure 1 plots the average number of flipped cards per trial in the hot and cold CCT versions as a function of potential gains, potential losses, and penalty cards. Flipped cards in the hot and cold CCT versions were the dependent variables in three mixed model analyses with each task design factor as a within-subjects factor. In all analyses, task type (i.e., hot vs. cold) was also set as a within-subject factor. Sex was a between-subject factor, age a covariate, and participant a random effect. Age was never a significant factor that affected risk choices in all analyses. Conversely, sex differences were always found. For example, the first analysis showed significant task [F(1,503) = 279.06; *p* < 0.001], gender [F(1,101) = 4.07; *p* < 0.05], and gender × task effects [F(1,922) = 5.91; *p* < 0.05].

Given the orthogonal design of task factors, these results were replicated in the subsequent analyses (Figure 2, panels a and b, respectively). Because increasing potential gains did not increase the average number of flipped cards (Figure 1, Panel a), adolescents did not consider this element in their risky decisions. Although the increase in potential losses tended to decrease the average number of cards turned in the cold CCT (Figure 1, Panel b), this factor was not significant, and therefore the adolescents did not consider it decisive information in refraining from turning cards. By contrast, the number of penalty cards on the display affected adolescents’ risky decisions [F(2,499) = 2.89; *p* = 0.06]. As shown in Figure 1 (Panel c), the number of flipped cards dropped substantially in the transition between 1 and 3 (t = −2.39; df = 499; *p* < 0.05). Collectively, these results showed that adolescents adopted a very simplified strategy during the game, not considering the available information except the number of penalty cards.

### 3.2. Network Analysis

Figure 3 shows the network plot including YSR, DERS, and CCT scores. With 16 nodes and 55/120 nonzero edges, the overall density of the network was 45.80%.

A visual inspection revealed two characteristics. First, YSR syndrome scales and difficulties in emotion regulation formed two ostensibly separated clusters. Second, there was some heterogeneity within each cluster. Regarding syndrome scales, stronger links were found among Anxious/Depressed, Withdrawal, and Somatization, reflecting the broad “internalizing” dimension of psychopathology. Likewise, Aggression and Rule Breaking, the two “externalizing” psychopathology scales, were also strongly interconnected. Social, Attention, and Thought problems were linked with both internalizing and externalizing scales. Regarding difficulties in emotion regulation, Awareness and Clarity were strongly interconnected, yet separated from Goals, Impulse, Nonacceptance, and Strategies. On the other hand, Goals, Impulse, Nonacceptance, and Strategies formed a visually interconnected group of variables that suggests difficulties in regulating emotions primarily at behavioral levels (e.g., problems behaving according to desired goals, difficulty in controlling impulsive behaviors, and limited access to effective strategies).

Centrality indices suggest which variables are in key positions in the network. Eight YSR syndrome scales ranked in the top ten in order of strength centrality, with Thought Problems, and Anxiety/Depression in the top two (Figure 4, Panel a, Strength Centrality). This finding reflects the density of the YSR cluster (Figure 1) and is consistent with the correlations among YSR syndrome scales reported in Table 1. Difficulties in emotion regulation were in the mid-range of the strength centrality list. Limited access to emotion regulation strategies was the strongest node in its cluster, ranking third by strength. The risk tasks were at the bottom of the list, reflecting the limited number of edges connecting them with other nodes in the network. As shown in Figure 4 (Panel b, Betweenness Centrality), the Thought Problems scale was the top node in terms of interconnections between nodes. As noted above, Thought Problems tied internalizing with externalizing scales and bridged YSR syndrome scales with difficulties in emotion regulation (especially Strategies and Nonacceptance) and risk propensity in the Cold CCT. The Strategy scale was also important in terms of betweenness centrality, given its intermediate position between difficulties in emotion regulation, YSR syndromes, and performance in the cold CCT. Noteworthy, the cold CCT was in the top ten list, ranking seventh by betweenness centrality, just after Anxiety/Depression and Social Problems. As shown in Figure 3, the cold CCT bridged Thought Problems, Strategies, and Clarity with performance in the hot CCT. Thought Problems were also the closest to all other nodes (Figure 4, panel c, Closeness Centrality). The Risk tasks were the nodes on average more distant from others. These results reflected the relative greater proximity of self-report scales (YSR and DERS) and the relative remoteness of the behavioral tasks from these interconnected clusters.

Although the EBIC-Glasso estimation yielded a parsimonious network structure, the network stability (CS = 0.20) was insufficient according to current standards (i.e., CS > 0.25). A low sample size relative to the number of nonzero-edges can produce instability. With this in mind, we redefined the network using a more parsimonious set of variables. In so doing, maladaptive psychological functioning and difficulties in emotion regulation were aggregated into new summary indices based on the theoretical structure of the YSR and DERS and empirical findings from the network plot (Figure 3). Thus, Anxiety/Depressed, Somatization, and Withdrawal were aggregated to obtain an Internalizing Problems (IP) index; Aggression and Rule-Breaking were collapsed into an Externalizing Problems (EP) index; Attention, Social Problems, and Thought problems made up an index of Mixed Cognitive Problems (MCP). Regarding the DERS, Goals, Impulse, Nonacceptance, and Strategies were used to define Difficulties in Regulating Emotions (DRE), while Awareness and Clarity defined Difficulties in Identifying Emotions (DIE). Not only Awareness and Clarity were originally thought to belong to the same construct reflecting a limited understanding of emotions and emotional expressivity [49] but also subsequent studies (e.g., [53]) showed that Awareness and Clarity items share a common factor, labeled Identification.

The redefined network model, with 7 nodes, 14/21 nonzero edges, and an overall density of 66.60%, was fairly stable (CS = 0.44). Analogue to Figure 3, Figure 5 showed that the MCP score was linked with both IP, EP, DRE, and Risk Propensity in the Cold CCT. As a result, MCP was the top node in the network for all centrality indices (Figure 6). Ranked second and third by strength, IP and EP were among the least important nodes in terms of Betweenness. While strength centrality represents the likelihood that a given node can activate other nodes in the network, betweenness can be interpreted as the degree to which a node can influence non-adjacent nodes, possibly explaining their associations. Accordingly, IP and EP can be thought of as highly influential in the network with regard to the activation of adjacent nodes, but not sufficiently “in the middle” of the plot to be able to explain the relationships between non-adjacent nodes. In contrast, MCP can be considered highly influential in terms of the ability to “mediate” between the other nonadjacent in the network. Similarly, the Cold CCT was found to be the second network node for betweenness centrality, bridging maladaptive psychological functioning and difficulties in emotion regulation with risk propensity in the Hot CCT. DRE and DIE were more peripheral in the network plot (Figure 5) but ranked higher in betweenness centrality than in strength (Figure 6). Their role in the network is indeed more nuanced than IP, EP, MCP, and CCT. By virtue of these indications, DRE and DIE may better fit the role of mediators between variables rather than exogenous variables.

### 3.3. Mediation Analyses

Integrating previous findings with the literature reviewed in the introduction, we hypothesized that IP and EP could be associated with risk propensity in the CCT through MCP, DRE, and DIE. Reproducing Figure 5, we also hypothesized that maladaptive psychological functioning and emotion regulation difficulties could be associated with risk propensity in Hot CCT through Cold CCT.

As shown in Figure 7, the model’s fit was excellent. The model explained 6% and 15% of the variance in hot and cold risky decisions, respectively. Consistent with the network analysis (Figure 5), the paths linking MCP with the Cold CCT and the latter with the Hot CCT were statistically significant, as was the corresponding indirect effect (IE 1 in Table 2). Given the strong significant paths from EP and IP to MCP (Figure 5 and Figure 7), the indirect effects from EP and IP to Hot CCT through MCP and Cold CCT were also significant (IE 4 and IE 7 in Table 2). Finally, DIE was also associated with Hot CCT through Cold CCT (IE 2 in Table 2).

## 4. Discussion

Our main hypothesis was that increased risk-taking would be associated with higher levels of maladaptive psychological functioning. Second, we expected that difficulties in emotion regulation would be associated with maladaptive psychological functioning and increased risk-taking. Third, we examined whether maladaptive psychological functioning and emotion regulation difficulties were differentially associated with risk propensity under deliberative and affective conditions. The first hypothesis was overall confirmed. The YSR syndrome scales were significantly associated with risk propensity in bivariate correlation analyses (especially in the deliberative condition), and the Mixed Cognitive Problem index was found to be central in the network structure and mediation analyses as a predictor of risk propensity. The second hypothesis was partially confirmed. On the one hand, difficulties in emotion regulation were associated with maladaptive psychological functioning. However, only difficulties in identifying emotions (especially clarity) were found to be consistently linked to risk propensity and to play a role in the network structure and mediation analyses. The third hypothesis concerned possible differences in the correlations between variables as a function of deliberative and affective conditions. In this regard, we found that cognitive problems, difficulties in identifying emotions, and risk propensity in the deliberative task formed the strongest associations in our study.

Before discussing the implications of our results, let us look at them in more detail. First, the YSR Thought Problems scale was the top node in the initial network structure for strength, betweenness, and closeness centrality, with a lack of emotional awareness, externalizing and internalizing syndrome scales, and risk propensity converging on it. Similarly, the Mixed Cognitive Problems index, with Thought Problems at its core, was found to be the most influential node in the redefined network structure. Although thought problems refer to strange perceptions, odd beliefs, and unusual behaviors, often associated with the positive symptoms of schizophrenia, psychotic-like experiences appear in several mental disorders [54] and are not uncommon in adolescents (e.g., up to 7.5% of adolescents between 13 and 18 years of age) [55]. Similarly, a recent Italian study estimated the prevalence of YSR Thought Problems to be approximately 6% in a large community sample [56], while another study showed the high diffusion of psychotic-like experiences in the Italian high-school population [57]. Considering the epidemiological data, our results suggest that adolescents with typical development may have transient difficulties with reality testing and disorganized thoughts, make errors in judgment, overlook probabilities and outcome information, and ultimately exhibit a high propensity for risk.

A second element supporting the role of cognitive factors emerged from the finding that difficulties in identifying emotions (i.e., DERS awareness and clarity) were linked with risk propensity in both CCT versions. Unlike other difficulties in regulating regulation based on behavioral strategies, awareness and clarity reflect a cognitive limitation in understanding and labeling emotions [49,53]. Previous research [58] has concluded that people lacking emotional clarity engage in high-arousal risk-taking behaviors (e.g., high-risk sports) because these activities offer easily identifiable elementary emotions (e.g., fear and joy). Because flipping cards in the hot CCT conveys a strong sense of escalating tension and excitement [40], the availability of high arousal and easily identifiable emotions may have made flipping cards more attractive to adolescents who lacked emotional awareness and clarity. However, this account seems insufficient to justify the same result obtained for the cold CCT. Other studies have shown that difficulty identifying feelings also predicts low arousal ethical risks (e.g., irresponsible academic and work behaviors) and aggressive behaviors in which the element of fun is absent [59]. In this perspective, risk-taking itself was thought to regulate unspecific negative affective states or escape from unclear emotional situations. Although cold CCT does not trigger any emotional arousal, it is not devoid of emotion [45]. Therefore, we believe the associations of difficulties identifying emotions with cold CCT could also arise because adolescents who cannot identify and describe integral emotions in affective decision-making may also fail to identify and use anticipated emotions in deliberative decision-making, that is a concept known as the affect-as-information heuristic [46].

The high betweenness centrality of the cold CCT in the network structure is the third important finding. Previous research has shown that risk propensity and use of information in the cold CCT are intertwined with cold executive functions [40,43,44]. A recent longitudinal study [35] found that the initial level of internalizing and externalizing problems predicted lower performance on cold executive function tasks four years later. In a similar vein, our study showed that most YSR syndrome scales were associated with risk propensity in the cold CCT and the latter with risk propensity in the hot CCT. These findings and the extant literature suggest that impairment in cold executive functions might link adolescents’ maladaptive functioning with risk-taking in conditions of high emotional arousal. Cold executive functions are thought to be of minor importance in the hot CCT, but some correlation exists [40,43]. In keeping with this literature, our study suggests that adolescents suffering from psychological symptoms may take excessive risks under conditions of high emotional arousal because of their limited ability to think analytically and plan a conservative strategy.

The above findings were also supported by a mediation analysis using the composite YSR and DERS indices as “predictors” of risk propensity. The Mixed Cognitive Problems index and risk propensity in the cold CCT mediated the relationship of Internalizing and Externalizing problems with risk propensity in the hot task. Our interpretation of indirect effects (i.e., from maladaptive psychological functioning to affective risk-taking through deliberative risk-taking) is consistent with recent findings that psychological symptoms precede the onset of neurocognitive deficits [35,36] and with the high prevalence of high-arousal dangerous behaviors (e.g., drug use, risky sexual behavior) in clinical samples [60,61]. The mediation model also revealed that only Difficulties Identifying Emotions were associated with risk propensity in the cold task (directly) and the hot task (indirectly), while behavioral emotion regulation strategies were ineffective to prevent risk-taking in the present study. This finding reinforces the theoretical link between cognitive emotion regulation, deliberative decision-making, and risk-taking in emotionally salient contexts (e.g., [45,58]).

We now discuss some unexpected findings. First, we found no age-specific trend in risk propensity. According to developmental theories [12], we expected to observe a peak of risk-taking in middle adolescence (i.e., 15–16 years), at least in the hot CCT. Indeed, previous research [40,62] showed that teenagers flipped more cards than young adults in the hot CCT, but not in the cold CCT. However, to the best of our knowledge, no study has compared early, middle, and late adolescents using this task. For example, Steinberg [28] used the IGT to demonstrate an inverted U-shaped trend in reward-seeking behavior by age, increasing from prepuberty to mid-adolescence and decreasing in later years. However, the tasks used in the present study and Steinberg [28] do not allow us to make a comparison under the same experimental conditions. Therefore, while our results replicate the trend of stable risk propensity in cold CCT observed for other age groups [40,62], they can neither confirm nor deny the trends in the 13–20 age group in hot CCT due to the lack of comparable reference data.

Second, teenagers adopted an oversimplified strategy to complete the risk task. Of the three design factors, only the number of penalty cards in the display proved significant in predicting the number of cards turned over, especially in the cold version of the task, where a tendency (though not significant) to turn over fewer cards as potential losses increased also emerged. In the original CCT study, adolescents were found to use less information in the hot CCT than in the cold CCT [40], a finding confirmed also in undergraduate and adult samples [43], and in the present study. Basing one’s choice on one piece of information only, adolescents used heuristics rather than analytical reasoning to cope with the cold task. Using no information at all in the hot task revealed that emotional arousal suppressed even the more basic forms of heuristic reasoning. Probably adolescents relied on other approaches, or misperceptions of random sequential events, such as the belief that if one has a run of successes/failures, one is more/less likely to have more success/failures in the future (e.g., a gambler’s fallacy) or irrational beliefs that unrelated random events are causally connected (e.g., superstition). These cognitive biases were associated with a greater risk of gambling and pathological gambling among Italian adolescents [63], real-life risky behaviors in which emotional arousal is a central component. Unfortunately, we did not expect our teenagers to misuse or not use the available information, and therefore we did not include independent measures of these cognitive biases.

Another unexpected finding was no differences in risk propensity between girls and boys in the hot CCT, whereas in the cold CCT girls flipped more cards than boys. According to Figner and Weber [64], there is strong evidence that males usually take more risks than females in laboratory tasks as well as in everyday situations (e.g., financial decisions). Regarding the hot CCT, undergraduate women were found to flip fewer cards than men [65]. However, a recent study found no gender differences in as many as four different risk tasks, including the cold CCT [66]. Unfortunately, our results are at odds with these recent studies, and we have no better explanation than to consider the age difference between our sample and the two references cited above.

Our study has important limitations, which may suggest avenues for future research. The sample is small and predominantly male. Even though our redefined network was sufficiently robust according to current standards [51], future research should attempt to cross-validate our results with a larger and more balanced sample. Second, the cross-sectional nature of the research does not allow us to interpret network and mediation analyses in a predictive or causal sense. Even though self-report scales and behavioral tasks were administered in two sessions spaced one week apart, adolescents took the cold and hot CCT in the same session. Therefore, to demonstrate that internalizing and externalizing symptoms and difficulties identifying emotions were linked with risk propensity in affective choices through deliberative processes, it would be better to test indirect effects using a longitudinal design in future research. Third, we did not include standard measures of cold executive functions as defined in previous research [32,35,36]. Because our results suggest that cognitive problems and executive functions are central to network structure, future research could expand the range of cognitive tests to isolate specific cognitive functions related to risk propensity in CCT.

Notwithstanding limitations, the results of the current study may have clinical implications for preventing risk-taking in youths. Several influential papers (e.g., [47,48]) maintain that network analysis can suggest primary targets for psychological interventions. For example, strength centrality is regarded as crucial because nodes with high strength centrality—Mixed Cognitive Problems and Internalizing Problems in our study—are thought to sustain the structure of the entire network. Thus, addressing these problems is thought to maximally impact the whole network. Betweenness centrality is also deemed important to detect nodes that bridge different variable clusters, such as maladaptive psychological functioning, difficulties in emotion regulation, and deliberative and affective risky decisions in teenagers in our study. These bridges can be viewed as maintaining the co-occurrence of different variable clusters in the network. Therefore, addressing the nodes with the highest betweenness centrality—Mixed Cognitive Problems and risk propensity in the Cold CCT in our study—might help to dispel the co-occurrence of risk propensity in affective decision contexts and both internalizing and externalizing syndromes. Finally, nodes with high closeness centrality—Mixed Cognitive Problems and Difficulties Regulating Emotions—are thought to have a more direct influence on the surrounding nodes. In sum, all centrality indices converge in suggesting Mixed Cognitive Problems as the primary target of interventions to prevent risk behaviors in adolescence. Breaking down the index of mixed cognitive problems into its constituent elements, it follows that attentional, social, and thinking problems are potential targets for psychological interventions aimed at preventing maladaptive psychological functioning from resulting in risky decisions with potentially harmful consequences. In a similar vein, several rehabilitation programs for children and adolescents can be found in the literature that aim to train executive functions to meet emotional and social needs by delaying impulsive decisions, improving problem-solving, and promoting flexible adaptation to changing priorities [67].

## 5. Conclusions

Adolescent risk-taking is complex and multidetermined. On the one hand, it represents a manifestation of maladaptive psychological functioning, and on the other hand, it depends on neurocognitive functions that reach full maturity only in young adulthood. Using network analysis as a heuristic tool to visualize the relationships among eight syndrome scales, six emotion regulation difficulties, and two behavioral tasks that hinge upon affective and deliberative decision processes helped us to address this complexity. We found that internalizing and externalizing syndromes and difficulties identifying emotions were linked with risk propensity in affective risky decisions through deliberative processes. This conclusion was further supported by statistically significant indirect effects in mediation analyses. Thus, our study highlights the role of cognitive factors in each variable set that might account for risk-taking in teenagers.

## Figures and Tables

**Figure 1 children-09-01915-f001:**
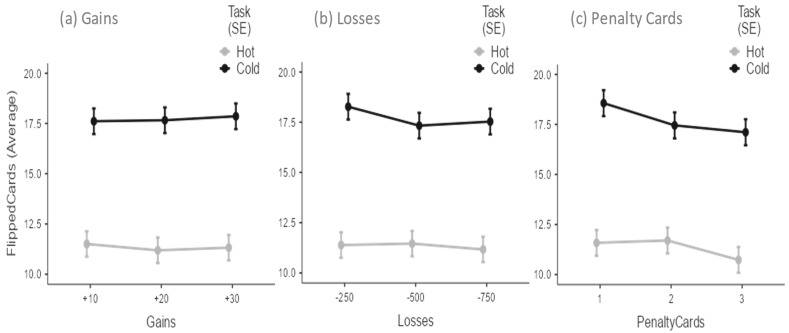
Average number of flipped cards per trial in the Hot and Cold Columbia Card Task as a function of (**a**) potential gains, (**b**) potential losses, and (**c**) penalty cards.

**Figure 2 children-09-01915-f002:**
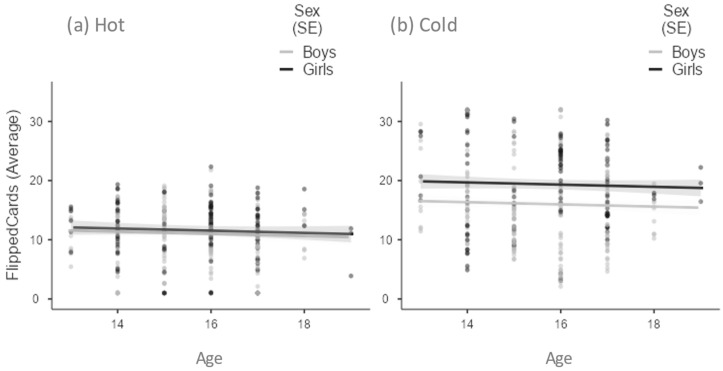
Average number of flipped cards per trial in the (**a**) Hot and (**b**) Cold Columbia Card Task as a function of age and gender.

**Figure 3 children-09-01915-f003:**
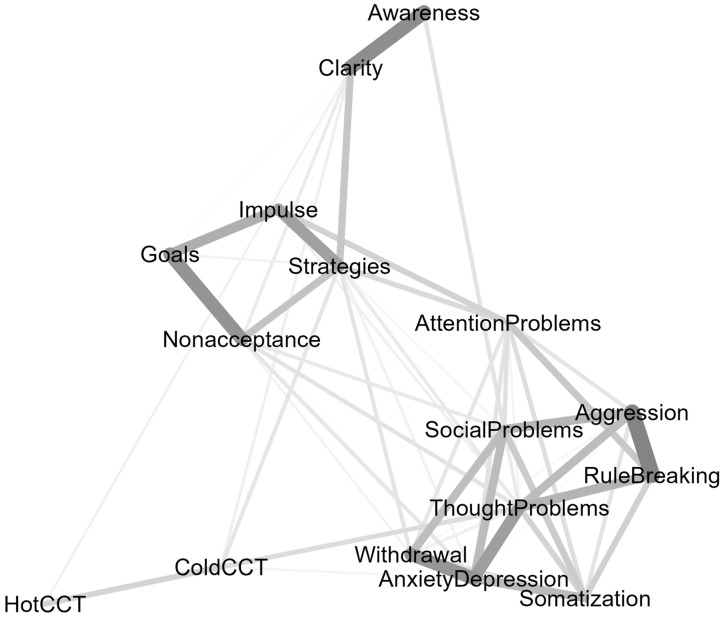
A network containing the eight syndrome scales of the Youth Self-Report, the six subscales of the Difficulties in Emotion Regulation Strategies, and risk propensity in decision-making tasks. The edge thickness represents the strength of an association.

**Figure 4 children-09-01915-f004:**
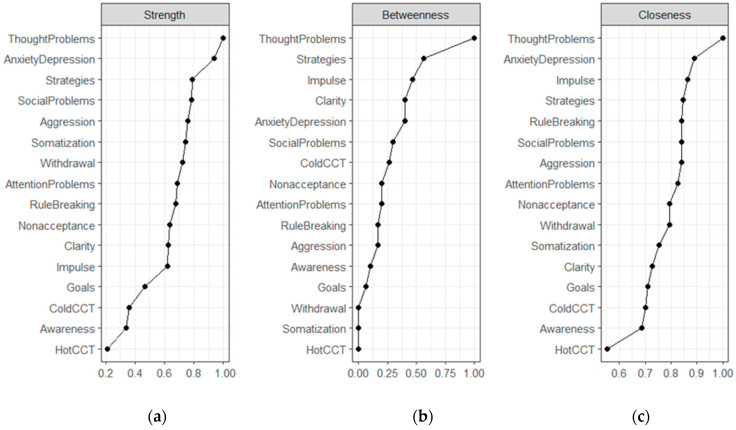
Node strength (**a**), betweenness (**b**), and closeness centrality (**c**) estimates for eight syndrome scales of the Youth Self-Report (YSR), six subscales of the Difficulties in Emotion Regulation Strategies (DERS), and risk propensity in the decision-making tasks.

**Figure 5 children-09-01915-f005:**
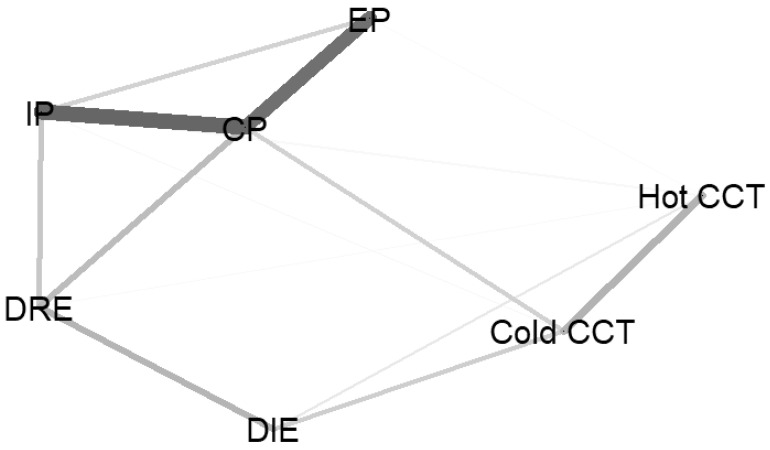
A network containing Externalizing Problems (EP), Internalizing Problems (IP), Cognitive Problems (CP), Difficulties in Regulating Emotions (DRE), Difficulties in Identifying Emotions (DIE), and risk propensity in the Hot and Cold CCT. The edge thickness represents the strength of an association. *Legend:* IP = Internalizing Problems; EP = Externalizing Problems; MCP = Mixed Cognitive Problems; DRE = Difficulties in Regulating Emotions; DIE = Difficulties in Identifying Emotions.

**Figure 6 children-09-01915-f006:**
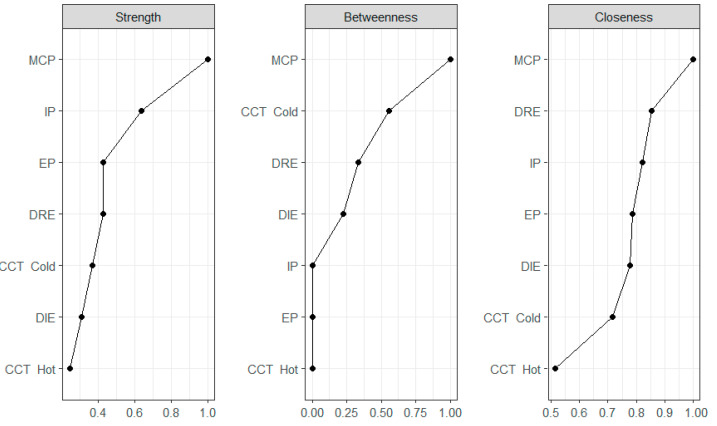
Node strength, betweenness, and closeness centrality estimates for three Youth Self-Report (YSR) indexes, two Difficulties in Emotion Regulation Strategies (DERS) indexes, and risk propensity in decision-making tasks.

**Figure 7 children-09-01915-f007:**
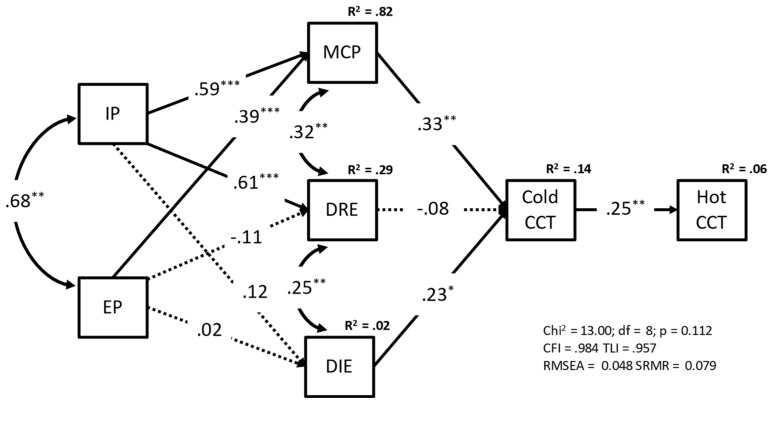
Path model in which maladaptive psychological functioning and difficulties in emotion regulation predicted deliberative and affective risky decisions in teenagers. Note: Standardized estimates are in the picture. Rectangles represent observed variables. Unidirectional straight arrows represent the regression paths. Bidirectional curved arrows represent covariance. Dotted lines indicate nonsignificant model parameters paths. Legend: IP = Internalizing Problems; EP = Externalizing Problems; MCP = Mixed Cognitive Problems; DRE = Difficulties in Regulating Emotions; DIE = Difficulties in Identifying Emotions. ** *p*< 0.01; *p* = *p*-value; df = degrees of freedom; Chi^2^ = Model’s chi square; TLI = Tucker-Lewis fit index; CFI = Comparative Fit Index; RMSEA = Root Mean Square Error of Approximation; SRMR = Squared Root Mean Square Residual; R^2^ = R squared.

**Table 1 children-09-01915-t001:** Intercorrelations among six subscales of the Difficulties in Emotion Regulation Strategies, eight syndrome scales of the Youth Self-Report, and two decision-making tasks.

Variables	Correlation Coefficients															
Nonacceptance	---	0.52 **	0.52 **	0.38 **	0.33 **	0.11	0.44 **	0.43 **	0.25 *	0.39 **	0.46 **	0.21 *	0.14	0.15	0.05	0.14
2. Goals	0.54 **	---	0.32 **	0.44 **	0.28 **	0.02	0.28 **	0.27 **	0.09	0.31 **	0.20*	0.25*	0.09	0.11	0.04	−0.06
3. Strategies	0.49 **	0.39 **	---	0.49 **	0.44 **	0.25 *	0.47 **	0.44 **	0.36 **	0.39 **	0.49 **	0.42 **	0.25 *	0.29 **	0.11	0.24 *
4. Impulse	0.35 **	0.47 **	0.55 **	---	0.24 *	0.14	0.37 **	0.26 **	0.26 **	0.45 **	0.50 **	0.45 **	0.37 **	0.34 **	−0.09	0.05
5. Clarity	0.30 **	0.24 *	0.38 **	0.26 **	---	0.49 **	0.19	0.19	0.02	0.23 *	0.20 *	0.16	0.03	0.00	0.20 *	0.27 **
6. Awareness	0.11	0.01	0.24 *	0.15	0.50 **	---	0.01	0.08	0.11	0.24 *	0.11	0.16	0.20 *	0.17	0.04	0.17
7. Anxious/Depressed	0.48 **	0.33 **	0.48 **	0.35 **	0.14	0.05	---	0.74 **	0.67 **	0.63 **	0.72 **	0.57 **	0.44 **	0.57 **	−0.10	0.24 *
8. Withdrawal	0.42 **	0.29 **	0.46 **	0.19	0.15	0.08	0.75 **	---	0.62 **	0.63 **	0.59 **	0.52 **	0.38 **	0.49 **	−0.07	0.18
9. Somatization	0.33 **	0.16	0.39 **	0.23 *	0.03	0.14	0.71 **	0.61 **	---	0.60 **	0.61 **	0.52 **	0.54 **	0.61 **	−0.15	0.10
10. Social Problems	0.45 **	0.22 *	0.44 **	0.38 **	0.16	0.29 **	0.73 **	0.67 **	0.67 **	---	0.54 **	0.54 **	0.47 **	0.57 **	−0.14	0.17
11. Thought Problems	0.47 **	0.21 *	0.49 **	0.41 **	0.14	0.15	0.78 **	0.63 **	0.69 **	0.67 **	---	0.58 **	0.64 **	0.66 **	0.05	0.28 **
12. Attention Problems	0.23 *	0.23 *	0.47 **	0.44 **	0.13	0.17	0.59 **	0.54 **	0.57 **	0.57 **	0.58 **	---	0.59 **	0.54 **	−0.07	0.19 *
13. Rule Breaking	0.15	−0.06	0.32 **	0.28 **	0.01	0.18	0.53 **	0.45 **	0.63 **	0.57 **	0.71 **	0.58 **	---	0.74 **	−0.07	0.16
14. Aggression	0.21 *	0.11	0.36 **	0.30 **	−0.01	0.23 *	0.63 **	0.56 **	0.63 **	0.68 **	0.72 **	0.57 **	0.77 **	---	−0.12	0.12
15. Hot CCT	0.05	0.09	0.14	−0.04	0.20 *	0.05	−0.03	−0.08	−0.09	−0.05	0.05	−0.01	−0.02	−0.09	---	0.33
16. Cold CCT	0.13	0.01	0.30 **	0.06	0.24 *	0.16	0.31 **	0.22 *	0.16	0.25 *	0.35 **	0.24 *	0.21 *	0.19	0.27 **	---
	1.	2.	3.	4.	5.	6.	7.	8.	9.	10.	11.	12.	13.	14.	15.	16.
M	12.83	12.33	19.42	13.07	12.10	7.52	7.12	4.26	3.90	4.20	5.47	5.44	6.34	8.35	11.28	17.23
SD	5.89	4.36	5.65	4.79	4.30	3.01	4.96	3.22	3.43	4.06	4.72	3.26	5.11	5.48	4.24	7.34

Note: Pearson r correlations below the diagonal, Spearman Rho correlations above the diagonal; **. Correlation is significant at the 0.01 level (two-tailed); *. Correlation is significant at the 0.05 level (two-tailed); N listwise = 100.

**Table 2 children-09-01915-t002:** Mediation Analyses: Test of indirect effects.

Label	Indirect Effect	Estimate	SE	Lower	Upper	β
IE1	MCP ⇒ Cold CCT ⇒ Hot CCT	0.03	0.02	0.01	0.07	0.08
IE2	DIE ⇒ Cold CCT ⇒ Hot CCT	0.04	0.02	0.01	0.10	0.06
IE3	DRE ⇒ Cold CCT ⇒ Hot CCT	−0.01	0.01	−0.03	0.01	−0.02
IE4	EP ⇒ MCP ⇒ Cold CCT ⇒ Hot CCT	0.01	0.01	0.00	0.03	0.03
IE5	EP ⇒ DRE ⇒ Cold CCT ⇒ Hot CCT	0.00	0.00	−0.00	0.01	0.00
IE6	EP ⇒ DIE ⇒ Cold CCT ⇒ Hot CCT	0.00	0.00	−0.01	0.01	0.00
IE7	IP ⇒ MCP ⇒ Cold CCT ⇒ Hot CCT	0.02	0.01	0.00	0.04	0.05
IE8	IP ⇒ DRE ⇒ Cold CCT ⇒ Hot CCT	−0.01	0.01	−0.03	0.01	−.01
IE9	IP ⇒ DIE ⇒ Cold CCT ⇒ Hot CCT	0.00	0.00	0.00	0.02	0.01

*Legend:* IP = Internalizing Problems; EP = Externalizing Problems; MCP = Mixed Cognitive Problems; DRE = Difficulties in Regulating Emotions; DIE = Difficulties in Identifying Emotions. Estimate = Unstandardized Indirect Effect; SE = Standard Error; Lower = Lower Limit Bootstrap Confidence Interval; Upper = Lower Limit Bootstrap Confidence Interval; β = Standardized Indirect Effect.

## Data Availability

All data generated or analyzed during this study are included in this published article as electronic Appendix A.

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
