# Peer review of "Deliberative and Affective Risky Decisions in Teenagers: Different Associations with Maladaptive Psychological Functioning and Difficulties in Emotion Regulation?"

_children, 2022, doi:10.3390/children9121915_

Round 1

Reviewer 1 Report

Hello

I think that overall this is a good study with no outstanding corrections required in my opinion - more just recommendations for a future study - please see report.

Many thanks

Author Response

Hello

I think that overall this is a good study with no outstanding corrections required in my opinion - more just recommendations for a future study - please see report.

Many thanks

Response: Thank You. We added recommendations for future studies in the discussion.

Reviewer 2 Report

In this work, the authors propose to investigate the relationships among three sets of variables that the literature suggests may underlie the surge in risk-taking during adolescence. In general, the manuscript is well written. The are some issues that must be addressed by the authors.

1.      Introduction. Page 3 line 116. The authors mention that “to the best of our knowledge, no study has made fine-grained comparisons between early, middle, and late adolescents using this task.” It is not clear, why this fact should be important in research. Please describe what is the main contribution of this work.

2.      Discussion. Please include a discussion of the implications of the results and how they can be used.

3.      Please include a conclusion section. There is not clear if the objectives are achieved. 

Author Response

In this work, the authors propose to investigate the relationships among three sets of variables that the literature suggests may underlie the surge in risk-taking during adolescence. In general, the manuscript is well written. The are some issues that must be addressed by the authors.

  1. Page 3 line 116. The authors mention that “to the best of our knowledge, no study has made fine-grained comparisons between early, middle, and late adolescents using this task.” It is not clear, why this fact should be important in research. Please describe what is the main contribution of this work.

Response: Thank You for your careful reading of our paper. Indeed, the sentence you reported to us may be misleading in that part of the manuscript. Following your recommendations, we have deleted that sentence and the related paragraph. The same concepts were already present in the discussion of our results, and it did not make sense to anticipate them out of context as you rightly pointed out.

  1. Please include a discussion of the implications of the results and how they can be used.

Response: In the revised manuscript we discussed the implications of our findings and how they can be used.

  1. Please include a conclusion section. There is not clear if the objectives are achieved.

Response: Thank you for pointing this out. Given the importance of clearly communicating whether the objectives were met, we clarified in the revision of the manuscript whether the study hypotheses were fully confirmed or not. We also added a general conclusion as recommended.

Reviewer 3 Report

The authors have a great work however they should address minor comments:

- improve the methods section by giving more details on:

    - participants characteristics;

    - procedure.

- clarify and improve the discussion section.

Author Response

The authors have a great work however they should address minor comments:

- improve the methods section by giving more details on:

    - participants characteristics; household

Response: Based on the data collected, we were able to provide information about ethnicity, household income, and whether or not the adolescents' family was intact.

    - procedure.

Response: We are unsure to which specific procedural aspects you were referring. We reviewed the procedure and we think we reported all information we had collected during the study regarding the procedure.

- clarify and improve the discussion section.

Response: The discussion has been thoroughly revised. In the revised version, we clarified whether the study hypotheses were fully confirmed or not, added recommendations for future studies, and discussed the implications of our findings and how they can be used.

Reviewer 4 Report

Thank you for your paper named "Deliberative and Affective Risky Decisions in Teenagers: Different Associations With Maladaptive Psychological Functioning and Difficulties in Emotion Regulation?".

It nicely addresses the issue regarding the role of neurological factors in the potential surge of risk-taking among teenagers.

Author Response

Thank you for your paper named "Deliberative and Affective Risky Decisions in Teenagers: Different Associations With Maladaptive Psychological Functioning and Difficulties in Emotion Regulation?".

It nicely addresses the issue regarding the role of neurological factors in the potential surge of risk-taking among teenagers.

Response: Thank you.